# Hybrid Knowledge Routed Modules for Large-scale Object Detection

**Chenhan Jiang**[*]
Sun Yat-Sen University
jchcyan@gmail.com

**Hang Xu**[*]
Huawei Noah's Ark Lab
xbjxh@live.com

**Xiaodan Liang**[†]
School of Intelligent Systems Engineering
Sun Yat-Sen University
xdliang328@gmail.com

**Liang Lin**
Sun Yat-Sen University
linliang@ieee.org

## Abstract

The dominant object detection approaches treat the recognition of each region separately and overlook crucial semantic correlations between objects in one scene. This paradigm leads to substantial performance drop when facing heavy long-tail problems, where very few samples are available for rare classes and plenty of confusing categories exists. We exploit diverse human commonsense knowledge for reasoning over large-scale object categories and reaching semantic coherency within one image. Particularly, we present Hybrid Knowledge Routed Modules (HKRM) that incorporates the reasoning routed by two kinds of knowledge forms: an explicit knowledge module for structured constraints that are summarized with linguistic knowledge (e.g. shared attributes, relationships) about concepts; and an implicit knowledge module that depicts some implicit constraints (e.g. common spatial layouts). By functioning over a region-to-region graph, both modules can be individualized and adapted to coordinate with visual patterns in each image, guided by specific knowledge forms. HKRM are light-weight, general-purpose and extensible by easily incorporating multiple knowledge to endow any detection networks the ability of global semantic reasoning. Experiments on large-scale object detection benchmarks show HKRM obtains around 34.5% improvement on VisualGenome (1000 categories) and 30.4% on ADE in terms of mAP. Codes and trained model can be found in https://github.com/chanyn/HKRM.

## 1 Introduction

The most state-of-the-art object detection methods [16, 43, 8, 4] follow the region-based paradigm, which treats the classification and boundingbox regression of each region proposal separately. The detection performance purely relies on the discriminative capabilities of region features, which often depends on sufficient training data for each category. Such paradigm thus obtains substantial performance drop when dealing with large-scale detection task [49, 18] that recognizes and localizes a large number of categories (e.g. 3000 classes in VG [23]). The long-tail problem is very common, where very few samples exist for rare classes, such as pepperoni and bagel. On the other hand, detection challenges such as heavy occlusion, class ambiguities and tiny-size objects become more severe due to more categories within one image. However, humans can still identity objects precisely under complex circumstances because of the remarkable reasoning ability resorting to commonsense

---

[*]Both authors contributed equally to this work.
[†]Corresponding Author

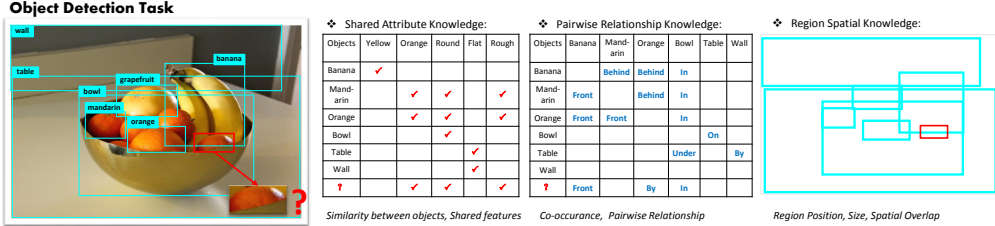

Figure 1: An example of how different types of commonsense knowledge can facilitate large-scale object detection, especially for rare classes (e.g. the obscured mandarin). We illustrate three useful knowledge forms: attribute knowledge, relationship knowledge and spatial knowledge.

knowledge. This inspires us to explore how to incorporate diverse knowledge forms into current detection paradigm in a light-weight and effective way, in order to mimic human reasoning procedure.

When humans watch a scene [3], each object is not identified individually. Different knowledge obtained by a human commonsense can help to make a correct identification by considering global semantic coherency. An example of hybrid knowledge reasoning in Figure 1 would be to identify the obscured "mandarin" (bottom-right). Human can recognize it is a mandarin learned from hybrid commonsense: a) this round object is orange and just like the other nearby mandarins (shared attribute knowledge); b) this object is in the bowl (pairwise relationship knowledge); c) this object has moderate size and its position is near to other fruits (spatial layout).

Recently, some works incorporate knowledge via direct relation modeling [34, 9, 19] or iterative reasoning architecture [33, 5, 6]. Different from recent implicit relation networks [19, 52] that learned inter-region relationships in an implicit and uncontrollable way, recently an iterative reasoning [6] was proposed to combine both local and global reasoning. However, they take only region predictions of a basic detection network, rather than enhancing intermediate feature representations. Furthermore, they directly use statistic edge connections in a prior knowledge graph while ignoring the compatibility of prior knowledge with visual evidence in each image. Given diverse object appearances and correlations in each image, personalized edge connections with respect to each knowledge form should be adaptive for different regions. On the contrary, our work aims to develop in-place knowledge modules which can not only explicitly incorporate any kinds of commonsense knowledge (both explicit or implicit) for better semantic reasoning but also link external knowledge with visual observations in each image in an adaptive way.

In this paper, we propose Hybrid Knowledge Routed Modules (HKRM) to incorporate multiple semantic reasoning routed by two major kinds of knowledge forms: an explicit knowledge module that exploits structure constraints that are summarized with linguistic knowledge (e.g. shared attributes, co-occurrence and relationships), and an implicit knowledge module to encode some implicit commonsense constraints over object (e.g. common spatial layouts). Instead of building category-to-category graph [26, 38, 22, 33, 7], each knowledge module in HKRM learns adaptive context connections for each pair of regions by regarding a specific prior knowledge graph as external supervisions, rather than fixing the connections. Our HKRM is general-purposed and extensible by easily integrating several individualized knowledge modules instantiated with any chosen knowledge forms to pursue more advanced and hybrid semantic reasoning. As a showcase, we experiment with three kinds of knowledge forms in this paper: the attribute knowledge (e.g. color, status), pairwise relationship knowledge such as co-occurrence and object-verb-subject relationship, the spatial knowledge including layout, size and overlap. HKRM is light-weight and easily plugged into any detection network for endowing its ability in global reasoning.

Our HKRM thus enables sharing visual features among certain regions with similar attributes, pairwise relationship or spatial relationship. The recognition and localization of difficult regions with heavy occlusions, class ambiguities and tiny-size problems can be thus remedied by discovering adaptive contexts from other regions guided by external knowledge. Another merit of HKRM lies in the ability of distilling common characteristics among common/uncommon categories so that the problem of crucial imbalanced categories can be alleviated.

The proposed HKRM outperforms the state-of-the-art Faster RCNN [43] with a large margin on two large-scale object detection benchmarks, that is, ADE [56] with 445 object classes and VG [23] with 1000 or 3000 classes. Particularly, our HKRM achieves around 34.5% of mAP improvement

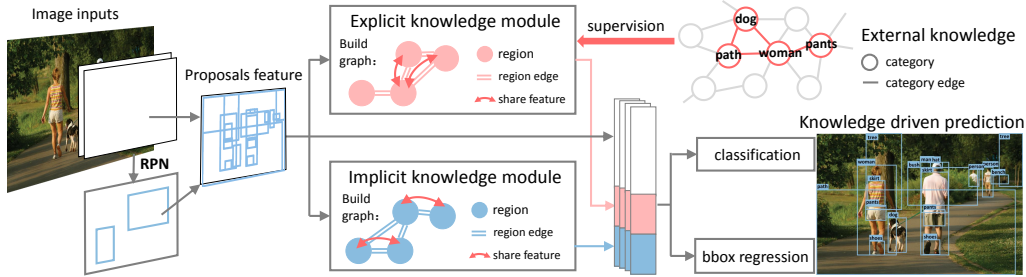

Figure 2: Overview of our HKRM, including two kinds of general modules: an explicit knowledge module to incorporate external knowledge and an implicit knowledge module to learn knowledge without explicit definitions or being summarized by human, such as spatial layouts. An adaptive region-to-region knowledge graph is constructed by regarding each specified external knowledge as the supervision of edge connections. The features of each region node are then enhanced through integrating several individual knowledge modules instantiated with distinct knowledge forms. The evolved features after each module are combined to produce final object detection results.

on VG (1000 categories), 26.5% on VG (3000 categories) and 30.4% on ADE. More interestingly, further analysis shows our HKRM module can provide meaningful explanations about how different commonsense knowledge can help perform reasonable visual reasoning and what each module actually learn with the guidance of external knowledge.

## 2    Related Work

**Object Detection.** Big progress has been made recent years on object detection due to the use of CNN such as Faster RCNN [43], R-FCN [8], SSD [30] and YOLO [41]. The backbones are some feature extractors such as VGG 16 [47] and Resnet 101 [17]. However, the number of categories being considered usually is small: 20 for PASCAL VOC [10] and 80 for COCO [29]. However, those methods are usually performed on each proposal individually without reasoning.

**Visual Reasoning.** Visual reasoning seeks to incorporate different information or interplay between objects or scenes. Several aspects such as shared attributes [11, 24, 39, 1, 2, 36], relationships among objects can be considered. [13, 32, 42] relies on finding similarity as the attributes in the linguistic space. For incorporating information such as relationship, most early works use object relations as a post-processing step [50, 14, 12, 37]. Recent works consider a graph structure [26, 38, 22, 33, 7, 6]. On the other hand, there are some sequential reasoning model for relationships [5, 25, 6]. In these works, a fixed graph is usually considered, while our module's graph has adaptive region-to-region edges which can be embedded with any kinds of external knowledge.

**Few-shot Recognition.** Few-shot recognition seeks to learn a new concept with a few annotated examples which share the similar problem with us. Early work focus on learning attributes embedding to represent categories [1, 21, 24, 44]. Most recent works use knowledge graph such as WordNet [35] to distill information among categories [46, 9, 54, 33, 53]. [15] further defined a GNN architecture to learn a knowledge graph implicitly. In contrast, our module is explicitly routed and benefits from the guidance of hybrid knowledge forms.

## 3    The Proposed Approach

### 3.1    Overview

The goal of this paper is to develop general modules for incorporating knowledge to facilitate large-scale object detection with global reasoning. Our HKRM includes two kinds of modules to support any prior knowledge forms, shown in Figure 2: an explicit knowledge module to incorporate external knowledge and an implicit knowledge module to learn knowledge without explicit definitions or being summarized by the human. Taking an image as the input, visual features are extracted for each proposal region through the region proposal network. Based on the region features, each module builds an adaptive region-to-region undirected graph $\hat{G}$ : $\hat{G} = < \mathcal{N}, \hat{\mathcal{E}} >$, where $\mathcal{N}$ are region

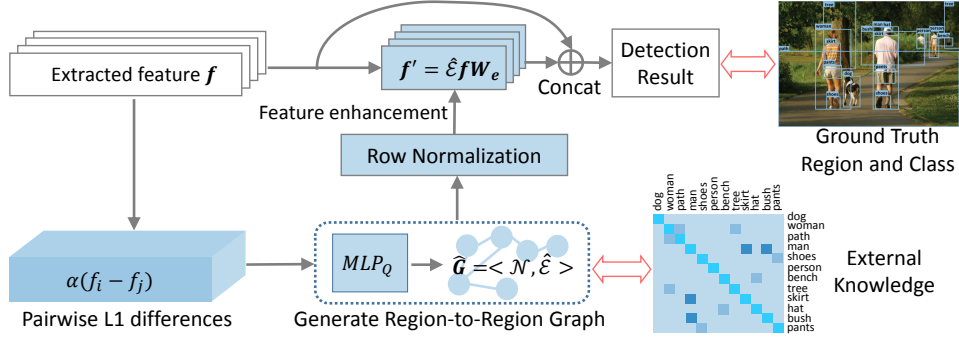

Figure 3: Explicit Knowledge Module. Taking the pairwise L1 differences of the **f** as inputs, a region-to-region graph is generated by stacked MLP. This process is supervised by the ground truth of the external knowledge. The output evolved feature **f′** is the enhanced feature via graph propagation. Then **f′** is concatenated to the **f** to produce final detection results.

proposal nodes and each edge $e_{i,j} \in \mathcal{E}$ defines a kind of knowledge between two nodes. Each module then outputs enhanced features integrating a particular knowledge. Finally, outputs from several modules are concatenated together and fed into the boundingbox regression layer and classification layer to obtain final detection results.

## 3.2 Explicit Knowledge Module

We regard the human commonsense knowledge that can be clearly defined and summarized using linguistics as *explicit knowledge*. The most representative explicit knowledge forms can be attribute knowledge (e.g. "apple is red.") and pairwise relationship knowledge (e.g. "man rides bicycles"). Our explicit knowledge module aims to enhance region features with kinds of explicit knowledge forms. Specifically, as shown in Figure 3, this module updates edge connections between each pair of region graph nodes in $\hat{G}$, supervised by a mapping of the ground truth from a class-to-class knowledge graph $Q$. This $Q$ is a certain form of linguistic knowledge.

### 3.2.1 Module Definition

**Adaptive region-to-region graph.** We first define a region-to-region graph $\hat{G}$ for all $N_r = |\mathcal{N}|$ region proposals with visual features $\mathbf{f} = \{f_i\}_{i=1}^{N_r}, f_i \in \mathbb{R}^D$ of $D$ dimension extracted from the backbone network, where $\mathcal{N}$ are region proposal nodes and $e_{i,j} \in \hat{\mathcal{E}}$ is the learned graph edge for each pair of region nodes. Given any external knowledge form, distinct edge connections $\hat{\mathcal{E}}$ can be accordingly updated to characterize specific context information for each region proposal in the context of specific knowledge. Formally, given a specific knowledge graph $Q$, each edge between two regions $\hat{e}_{ij}$ is learned by a stacked Multi-layer Perceptron (MLP) :

$$\hat{e}_{ij} = \text{MLP}_{\boldsymbol{Q}}(\alpha(f_i, f_j)), \tag{1}$$

where $\alpha(\cdot)$ is chosen to be the pairwise L1 difference between features of each region pair $(f_i, f_j)$ since L1 difference is symmetric. Given different prior graphs $Q$, $\text{MLP}_{\boldsymbol{Q}}$ would be parametrized with $W_{\boldsymbol{Q}}$ distinctly to generate different region-to-region graphs $\hat{G}$, leading to personalized knowledge reasoning.

We learn $\text{MLP}_{\boldsymbol{Q}}$ by directly enforcing the predicted $\hat{e}_{ij}$ to be consistent with the edge weights of a prior graph $Q$. We define $\boldsymbol{Q} =< \mathcal{C}, \mathcal{V} >$ as a class-to-class graph with $C$ class graph nodes and their prior edge weights $v_{i,j} \in \mathcal{V}$, such as attribute and relationship graphs. During training, as we know ground-truth categories of each region, the edge $\hat{e}_{ij}$ of two region nodes is learned towards the edge weights $\tilde{e}_{ij}$ of ground truth categories of region nodes in $Q$, that is, $\tilde{e}_{ij} = v_{c_i,c_j}$ where $c_i$ is the ground truth class of $i$-th region. Such explicit supervision with ground truth classes of region nodes would ensure the learning of a reliable graph reasoning regardless of the errors from proposal localization. $\text{MLP}_{\boldsymbol{Q}}$ is then learned to encode explicit region-wise knowledge correlations that can be applied in the testing phase. The loss function of learned edge weights $\{\hat{e}_{ij}\}$ for all $N_r$ region proposals is defined as:

$$\mathcal{L}(\mathbf{f}, W_{\boldsymbol{Q}}, \boldsymbol{Q}) = \sum_{i=1}^{N_r} \sum_{j=1}^{N_r} \frac{1}{2}(\hat{e}_{ij} - \tilde{e}_{ij})^2. \tag{2}$$

**Feature evolving via graph reasoning.** After performing row normalization over learned edges $\hat{\mathcal{E}} = \{\hat{e}_{ij}\}$, we can propagate features of connected regions into enhancing each region features $\mathbf{f}'$ by different weighted edges, which can be solved by matrix multiplication:

$$\mathbf{f}' = \hat{\mathcal{E}}\mathbf{f}\boldsymbol{W}_e, \tag{3}$$

where $\boldsymbol{W}_e \in \mathbb{R}^{D \times E}$ is a transformation weight matrix and $\mathbf{f}' \in \mathbb{R}^E$ are the enhanced features with $E$ dimension via graph reasoning. Those regions with heavy occlusions, class ambiguities and the tiny-size problem can be remedied by discovering adaptive contexts from other regions guided by external knowledge. The trainable parameters are $W_{\boldsymbol{Q}}$ of the stacked MLP and $\boldsymbol{W}_e$.

### 3.2.2 Module Specification with Different Knowledge

We can specify different prior knowledge graphs $\boldsymbol{Q}$ to obtain distinct graph reasoning behaviors. Here, we take attribute knowledge graph and relationship knowledge graph as the examples. We refer readers to find illustrations of constructing knowledge graphs in Supplementary material.

**Attribute Knowledge.** Attribute knowledge graph $\boldsymbol{Q^A}$ as one kind of $\boldsymbol{Q}$ denotes object classes are connected with kinds of attributes such as colors, size, materials, and status. The explicit knowledge module instantiated with attribute knowledge will facilitate features of rare classes with more frequent classes by transferring their shared visual attribute properties. Let us consider $C$ classes and $K$ attributes, we obtain a $C \times K$ frequency distribution table for each class-attribute pair, detailed in experiments. Then the pairwise Jensen–Shannon (JS) divergence between probability distributions $P_{c_i}$ and $P_{c_j}$ of two classes $c_i$ and $c_j$ can be measured as the edge weights of two classes $e^A_{c_i, c_j} = JS(P_{c_i} || P_{c_j})$. We consider JS divergence to measure the similarity instead of KL divergence here since we expect a symmetry undirected graph while $KL(P_i || P_j) \neq KL(P_j || P_i)$. Finally, the module outputs a enhanced feature $\mathbf{f}'_a \in \mathbb{R}^{E_a}$.

**Relationship Knowledge.** Relationship knowledge $\boldsymbol{Q^R}$ denotes the pairwise relationship between classes, such as location relationship (e.g. *along*, *on*), the "subject-verb-object" relationship (e.g. *eat*, *wear*) or co-occurrence relationship. The evolved features will be enhanced with high-level semantic correlations between regions. Similarly, we obtain $\boldsymbol{Q^R}$ by calculating frequent statistics either from the semantic information or simply from the occurrence among all class pairs. The symmetric transformation and row normalization are performed on edge weights. Let $\mathbf{f}'_r \in \mathbb{R}^{E_r}$ denotes the output of the explicit relationship module.

### 3.3 Implicit Knowledge Module

Considering some commonsense knowledge without explicit definitions or being summarized by the human, we regard them as *implicit knowledge* and thus an implicit knowledge module is designed. Taking geometry priors as an example, besides those explicit pairwise locations, there also exists some complicate location information, such as "the ceiling is always above all the other objects" and "the water is always below the ships, mountains and the sky". Taking features $\mathbf{q} = \{q_i\}$ as inputs that depict the features of each region (e.g. geometric features), our implicit knowledge module integrates multiple graph reasoning over $M$ region-to-region graphs obtained by M stacked MLPs following (1) to encode these implicit priors. The analogous idea of multi-head attention can be found in [6, 19, 51]. This enables the module to catch multiple spatial relationships such as "up and down", "left and right" and "corner and center". Visualization of different learned graphs can be found in Supplementary material. Similar to region-to-region graph used in explicit knowledge module, we learn specific edge weights $\{\hat{e}_{ij}^{(m)}\}$ of each graph $\hat{\mathbf{G}}_m, m = 1, \dots, M$ for all-region proposal pairs, following Eqn. 1. We then average edge weights of all graph $\{\hat{\mathbf{G}}_m\}$ and add them with a identity matrix $\boldsymbol{I}$ to obtain the edge connections $\hat{e}_{ij}^I \in \hat{\mathcal{E}}^I$:

$$\hat{e}_{ij}^I = \frac{1}{M} \sum_{m=1}^{M} \hat{e}_{ij}^{(m)} + \boldsymbol{I}. \tag{4}$$

| % | Method | AP | $AP_{50}$ | $AP_{75}$ | $AP_S$ | $AP_M$ | $AP_L$ | $AR_1$ | $AR_{10}$ | $AR_{100}$ | $AR_S$ | $AR_M$ | $AR_L$ |
|---|---|---|---|---|---|---|---|---|---|---|---|---|---|
| $VG_{1000}$ | Light-head rcnn[27] | 6.2 | 10.9 | 6.2 | 2.8 | 6.5 | 9.8 | 14.6 | 18.0 | 18.7 | 7.2 | 17.1 | 25.3 |
| | FPN[28] | 5.6 | 10.1 | 5.4 | 3.4 | 5.8 | 8.0 | 13.0 | 16.5 | 16.6 | 7.5 | 15.7 | 20.6 |
| | Faster RCNN[43] | 5.8 | 10.7 | 5.7 | 1.9 | 5.8 | 10.0 | 13.7 | 17.2 | 17.2 | 4.9 | 15.7 | 25.3 |
| | Attribute | 7.4 | 12.9 | 7.4 | 2.4 | 7.4 | 13.7 | 17.0 | 21.4 | 21.5 | 6.0 | 19.5 | 33.0 |
| | Relation | 7.4 | 12.8 | 7.5 | 3.0 | 7.5 | 13.0 | 17.0 | 21.6 | 21.7 | 7.2 | 19.8 | 31.4 |
| | Spatial | 7.3 | 12.1 | 7.7 | 2.7 | 7.2 | 12.7 | 17.7 | 21.9 | 22.0 | 6.3 | 19.5 | **33.3** |
| | HKRM (All) | $7.8^{+2.0}$ | $13.4^{+2.7}$ | $8.1^{+2.4}$ | $4.1^{+2.2}$ | $8.1^{+2.3}$ | $12.7^{+2.7}$ | $18.1^{+4.4}$ | $22.7^{+5.5}$ | $22.7^{+5.5}$ | $9.6^{+4.7}$ | $20.8^{+5.1}$ | $31.4^{+6.1}$ |
| $VG_{3000}$ | Light-head rcnn[27] | 3.0 | 5.1 | 3.2 | 1.7 | 4.0 | 5.8 | 7.3 | 9.0 | 9.0 | 4.3 | 10.3 | 15.4 |
| | FPN[28] | 3.3 | 5.2 | 3.2 | 1.9 | 4.3 | 4.8 | 6.9 | 8.3 | 8.3 | 4.3 | 9.8 | 11.6 |
| | Faster RCNN[43] | 3.4 | 6.0 | 3.4 | 1.6 | 4.3 | 7.3 | 8.1 | 9.8 | 9.8 | 3.8 | 10.9 | 17.0 |
| | Attribute | 4.1 | 7.0 | 4.3 | 2.5 | 5.3 | 7.9 | 9.7 | 11.7 | 11.7 | 5.7 | 12.8 | 19.6 |
| | Relation | 4.2 | 7.1 | 4.3 | 2.6 | 5.3 | 8.1 | 9.7 | 11.9 | 11.9 | 6.0 | 12.8 | 19.8 |
| | Spatial | 4.0 | 6.7 | 4.1 | 2.3 | 5.1 | 7.6 | 9.3 | 11.2 | 11.2 | 5.3 | 12.4 | 18.7 |
| | HKRM (All) | $4.3^{+0.9}$ | $7.2^{+1.2}$ | $4.4^{+1.0}$ | $2.6^{+1.0}$ | $5.5^{+1.2}$ | $8.4^{+1.1}$ | $10.1^{+2.0}$ | $12.2^{+2.4}$ | $12.2^{+2.4}$ | $5.9^{+2.1}$ | $13.0^{+2.1}$ | $20.5^{+2.5}$ |
| ADE | Light-head rcnn[27] | 7.0 | 11.7 | 7.3 | 2.4 | 5.1 | 11.2 | 9.6 | 13.3 | 13.4 | 4.3 | 10.4 | 20.4 |
| | FPN[28] | 6.5 | 12.1 | 6.2 | 3.3 | 6.0 | 10.5 | 9.5 | 12.9 | 13.0 | 5.3 | 11.9 | 18.6 |
| | Faster RCNN[43] | 7.9 | 14.7 | 7.5 | 2.1 | 5.8 | 13.2 | 10.6 | 14.2 | 14.4 | 4.5 | 11.9 | 22.4 |
| | Attribute | 9.6 | 16.8 | 9.7 | 3.1 | 7.0 | 15.9 | 12.7 | 16.9 | 17.1 | 6.1 | 14.1 | 26.3 |
| | Relation | 9.6 | 16.8 | 9.8 | 3.0 | 7.2 | 15.4 | 12.6 | 16.8 | 17.0 | 6.2 | 14.2 | 26.0 |
| | Spatial | 8.7 | 14.0 | 9.0 | 3.1 | 6.9 | 14.3 | 11.4 | 15.5 | 15.6 | 5.0 | 12.7 | 24.2 |
| | HKRM (All) | $\mathbf{10.3}^{+2.4}$ | $\mathbf{18.0}^{+3.0}$ | $\mathbf{10.4}^{+2.9}$ | $\mathbf{4.1}^{+2.0}$ | $\mathbf{7.9}^{+2.1}$ | $\mathbf{16.8}^{+3.6}$ | $\mathbf{13.6}^{+3.0}$ | $\mathbf{18.3}^{+4.1}$ | $\mathbf{18.5}^{+4.1}$ | $\mathbf{7.1}^{+2.6}$ | $\mathbf{15.5}^{+3.6}$ | $\mathbf{28.4}^{+6.0}$ |

Table 1: Main results of test datasets on $VG_{1000}$, $VG_{3000}$ and ADE. "Attribute", Relation" and "Spatial" are the baseline Faster RCNN adding the corresponding knowledge module alone. HKRM is the model with a combination of all.

We then adopt matrix multiplication $\mathbf{g}' = \hat{\mathcal{E}}^I \mathbf{f} \boldsymbol{W}_g$ to get the evolved features $\mathbf{g}' \in \mathbb{R}^{E_g}$. The trainable parameters are weights of $M$ stacked MLP for learning edge weights of knowledge graphs $\{\hat{\mathbf{G}}_m\}$, and the transformation matrix $\boldsymbol{W}_g \in \mathbb{R}^{D \times E_g}$ is shared for all graphs.

**Module specification with spatial layout.** Here, we instantiate the implicit knowledge module by spatial layout inputs to capture complicated spatial constraints by using specific input information. The input geometry feature $q_i$ of each region is simply object bounding box. To make $q_i$ be invariant to the scale transformation, a relative geometry feature is used, as $(\frac{x_i}{\bar{w}}, \frac{y_i}{\bar{h}}, \frac{w_i}{\bar{w}}, \frac{h_i}{\bar{h}}, p_i)$, where $\bar{w}$ and $\bar{h}$ denotes the size of the image and $p_i$ is the initial foreground probability of each region. Note that edge weights are implicitly learned via the back-propagation of the whole network.

## 4   Experiments

**Dataset and Evaluation.** We conduct experiments on large-scale object detection benchmarks with a large number of classes: that is, Visual Genome (VG) [23] and ADE [56]. The task is to localize an object and classify it, which is different from the experiments with given ground truth locations [6]. For Visual Genome, we use the latest release (v1.4), and synsets [45] instead of the raw names of the categories due to inconsistent label annotations, following [20, 6]. We consider two set of target classes: 1000 most frequent classes and 3000 most frequent classes, resulting in two settings $VG_{1000}$ and $VG_{3000}$. We split the remaining 92960 images with objects on these class sets into 87960 and 5,000 for training and testing, respectively. In term of ADE dataset, we use 20,197 images for training and 1,000 images for testing, following [6]. To validate the generalization capability of models and the usefulness of transferred knowledge graph from VG, 445 classes that overlap with VG dataset are selected as targets. Since ADE is a segmentation dataset, we convert segmentation masks to bounding boxes [6] for all instances. For evaluation, we adopt the metrics from COCO detection evaluation criteria [29], that is, mean Average Precision (mAP) across different IoU thresholds (IoU= $\{0.5 : 0.95, 0.5, 0.75\}$) and scales (small, medium, big). We also use Average Recall (AR) with different number of given detection per image ($\{1, 10, 100\}$) and different scales (small, medium, big).

Additionally, we also evaluate on PASCAL VOC 2007 [10] and MSCOCO 2017 [29] to show prior knowledge can help detection for a small set of frequent classes (20/80 classes). PASCAL VOC consists of about 10k trainval images (included VOC 2007 trainval and VOC 2012 trainval) and 5k

| Dataset | Method | Backbone | #. Parameter (M) | mAP (%) |
|---|---|---|---|---|
| PASCAL VOC$_{20}$ | SMN[5] | ResNet-101 | 66.7 | 67.8 |
| | Faster RCNN[43] | ResNet-101 | 57.0 | 75.1 |
| | HKRM (All) | ResNet-101 | 59.2 | **78.8** |
| MSCOCO$_{80}$ | SMN[5] | ResNet-101 | 68.1 | 31.6 |
| | Relation Network[19] | ResNet-101 | 64.6 | 35.2 |
| | Faster RCNN[43] | ResNet-101 | 58.3 | 34.2 |
| | HKRM (All) | ResNet-101 | 60.3 | **37.8** |

Table 2: Comparisons of mean Average Precision (mAP) and #. Parameter on PASCAL VOC 2007 test set and COCO 2017 val set.

test images over 20 object categories. We only report mAP scores using IoU thresholds at $0.5$ for the purpose of comparison with other existing methods. MSCOCO 2017 contains 118k images for training, 5k for evaluation.

**Knowledge Graph Construction.** We apply general knowledge graphs for both experiments on VG and ADE datasets. With the help of the statistics of the annotations in the VG dataset, we can both create attribute knowledge and relationship knowledge graphs. Specifically, we consider top 200 most frequent attributes annotations in VG such as color, material and status of the categories ($C = 3000$), and then count their frequent statistics as the class-attribute table. For relationship knowledge, we use top 200 most frequent relationship annotations in VG such as location relationship, subject-verb-object relationship, and count frequent statistics of each class-relationship pair. Illustrations of constructed knowledge graphs can be found in Supplementary material.

**Implementation Details.** We treat the state-of-the-art Faster RCNN [43, 55] as our baseline and implement all models in Pytorch [40]. We also compare with Light-head RCNN [27] and FPN [28]. ResNet-101 [17] pretrained on ImageNet [45] is used as our backbone network. The parameters before conv3 and the batch normalization are fixed, same with [27]. During training, we augment with flipped images and multi-scaling (pixel size=$\{400, 500, 600, 700, 800\}$). During testing, pixel size= $600$ is used. Following [43], RPN is applied on the conv4 feature maps. The total number of proposed regions after NMS is $128$. Features in conv5 are avg-pooled to become the input of the final classifier. Unless otherwise noted, settings are same for all experiments. In terms of our explicit attribute and relationship knowledge module upon region proposals, we use the final conv5 for 128 regions after avg-pool ($D= 2048$) as our module inputs. We consider a 4 stacked linear layers as $\text{MLP}_Q$(output channels:$[256, 128, 64, 1]$). ReLU is selected as the activation function between each linear layer. The output size : $E_a = E_r = 256$, which is considered sufficient to contain the enhanced feature. In terms of implicit knowledge module, we employ $M = 10$ implicit graphs. For learning each graph, 2 stacked linear layers are used (output channels:$[5, 1]$). $p_i$ is the score of the foreground form the RPN. The output size: $E_g = 256$. To avoid over-fitting, the final version of HKRM is the combination of three shrink modules with each output size equals 256. $\mathbf{f}'_a$, $\mathbf{f}'_r$, $\mathbf{g}'$ and $\mathbf{f}$ are concatenated together and fed into the boundingbox regression layer and classification layer. We apply stochastic gradient descent with momentum to optimize all models. The initial learning rate is 0.01, reduce three times ($\times 0.01$) during fine-tuning; $10^{-4}$ as weight decay; 0.9 as momentum. For both VG and ADE dataset, we train 28 epochs with mini-batch size of 2 for both the baseline Faster RCNN. (Further training after 14 epochs won't increase the performance of baseline.) For our HKRM, we use 14 epochs of the baseline as pretrained model and train another 14 epochs with same settings with baseline.

### 4.1 Comparison with state-of-the-art

We report the result comparisons on VG$_{1000}$ with 1000 categories , VG$_{3000}$ with 3000 categories and ADE dataset in Table 1. As can be seen, all our model variants outperform the baseline Faster RCNN[43] on all dataset. Our HKRM achieves an overall AP of 7.8% compared to 5.8% by Faster RCNN on VG$_{1000}$, 4.3% compared to 3.4% on VG$_{3000}$, and 10.3% compared to 7.9% on ADE, respectively. Moreover, our model achieves significant higher performance on both classification and localization accuracy than the baseline on all cases (i.e. different scales and overlaps). This verifies the effectiveness of incorporating global reasoning guided by rich external knowledge into local region recognition. More significant performance gap by our HKRM can be observed for those

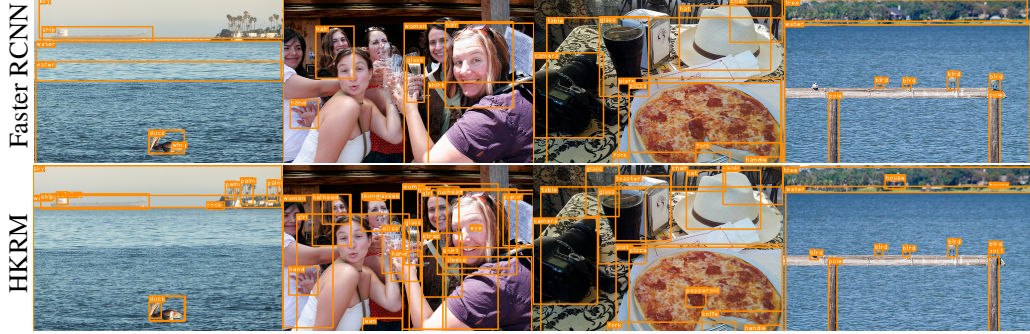

Figure 4: Qualitative result comparison on $VG_{1000}$ between Faster RCNN and our HKRM. Objects with occlusion, ambiguities and rare category can be detected by our modules.

rare categories with very few samples (about 1.5% average improvement for the top 150 infrequent categories by our method in terms of mAP).

To compare with the state-of-art knowledge-enhanced methods, we also implement HKRM on PASCAL VOC and MS COCO datasets with only 20/80 categories in Table 2. For PASCAL VOC, our HKRM performs 1.1% better than the baseline Faster RCNN, and outperforms Spatial Memory Network [5]. For MSCOCO, comparison is made between Relation Network [19] and Spatial Memory Network. The proposed HKRM boosts the mAP from 34.9% to 37.8% and outperform all the other methods. Our method can also boost the performance in the more simplified dataset benefiting from the shared linguistic knowledge and spatial layout knowledge. Note that HKRM consisted of three knowledge modules totally increases about 2% parameters and is light-weight compared to [5, 19].

Figure 4 shows the qualitative result comparison between our HKRM and Faster RCNN. Our HKRM can detect the obscure palm trees far away in the left image. In the middle image, the multiple overlapped small objects such as glass and paper is recognized by our method. "Pepperoni" is a rare category and is detected on the pizza in the right image.

## 4.2 Ablation Studies

**The effect of different explicit knowledge.** We analyze the effect of both attribute and relationship knowledge on final detection performance. The attribute module along can increase overall AP by 1.6% for $VG_{1000}$, 0.6% for $VG_{3000}$ and 1.7% for ADE over baseline. The relationship module has similar performance with a slightly higher result for $VG_{3000}$. Sharing visual feature according to both attribute and relationship knowledge can actually boost the performance of object detection.

**The effect of different explicit knowledge.** We analyze the effect of both attribute knowledge and relationship knowledge on final detection performance. The attribute module along can increase overall AP by 1.6% for $VG_{1000}$, 0.6% for $VG_{3000}$ and 1.7% for ADE over baseline. The relationship module has similar performance with a slightly higher result for $VG_{3000}$. Sharing visual feature according to both attribute and relationship knowledge can actually boost the performance of object detection.

**The effect of implicit knowledge.** As can be seen, the implicit spatial module alone helps around 1.5% for $VG_{1000}$, 0.3% for $VG_{3000}$ and 0.8% for ADE. The spatial module alone is not as effective 263 as the attribute and relation module. However, the unsupervised learning of the spatial knowledge 264 still can significantly help the object recognition through those undefined knowledge.

**Generalization capability.** From Table 1, the external knowledge graph from VG can actually help to improve the performance of ADE. Therefore, any datasets with overlap categories can share the existing knowledge graph. Besides, our module can be added to diverse detection systems easily.

**Global reasoning.** The proposed HKRM achieves the global reasoning over regions via one-time propagation over all graph edges and nodes. Benefiting from the learned knowledge graph for each image, our HKRM is able to propagate information between nodes which are not connected in the prior knowledge graph. We have tried the higher orders of feature transformation (e.g. 2 and 3) and did not observed significant improvement. In fact, over-transformation will even make the enhanced features all identical.

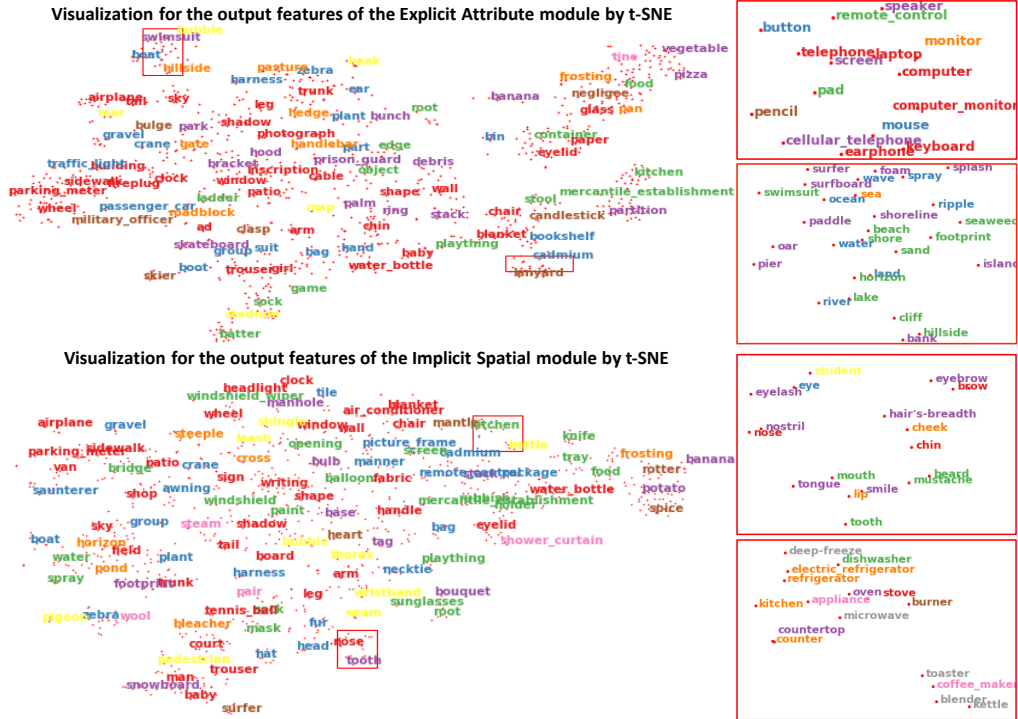

Figure 5: 2-D visualization of $\mathbf{f}'_a$ and $\mathbf{g}'$ by t-SNE method [31]: the explicit module with attribute knowledge (top); implicit knowledge module with spatial knowledge (bottom) . The red regions are enlarged in right panels. The categories shared the similar attribute knowledge (top) and spatial relationship (bottom) are closed to each other. This verifies that our modules learn the corresponding knowledge.

**Analysis of feature interpretability.** To better understand the underlying feature representations that our HKRM actually learn for graph reasoning, we record the output $\mathbf{f}'_a$ and $\mathbf{g}'$ ($E_a = E_g = 512$) from the explicit attribute module and implicit spatial module and its corresponding real labels from each region of 8000 $VG_{1000}$ images. Then we take average according to the labels and use the t-SNE [31] clustering method to visualize them as shown in Figure 5. Note that if features of some categories are closed to each other, the edges between those close categories are more likely to be activated. From two enlarged regions on top, we can see that features of categories which share similar attributes such as "water", "sand" and "electronics" are closed to each other. And this speaks well our explicit knowledge module successfully incorporates the prior attribute knowledge and leads to interpretable feature learning. Similarly, from two bottom enlarged regions, features of objects which has spatial relationship such as "on face" and "in kitchen counter" are closed to each other. This validates our spatial knowledge module is capable of encoding underlying spatial relationships. Benefiting from explicit knowledge supervision, the feature clustering property of the explicit attribute module seems to be better than those of the implicit knowledge module. More gradient visualization [48] results for the enhanced features are included in Supplementary materials for better understanding the module.

## 5 Conclusion

We present two novel general knowledge modules in HKRM. The first one can embed any external knowledge through supervision. The second one can implicitly learn some knowledge without explicit definitions or being summarized by human. Both modules can be easily applied to the original detection system to improve the detection performance. The experiment and analysis indicated HKRM can alleviate the problems of large-scale object detection. For our future work, we can use Cholesky decomposition to re-parametrize the region-to-region graph to further reduce half of the module parameters due to the property of symmetry of our graph. We can also add experiments using the word embedding knowledge in the explicit module and the latest new Open Images Dataset which consists about 600 categories.

**Acknowledgments**

This work was supported in part by the National Key Research and Development Program of China under Grant No. 2018YFC0830103, in part by National High Level Talents Special Support Plan (Ten Thousand Talents Program), and in part by National Natural Science Foundation of China (NSFC) under Grant No. 61622214, and 61836012.

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
