[Supplementary Material · Supplementary_Materials_Final.pdf]

# Supplementary materials
# Hybrid Knowledge Routed Modules for Large-scale Object Detection

**Chenhan Jiang**[*]
Sun Yat-Sen University
jchcyan@live.com

**Hang Xu**[*]
Huawei Noah's Ark Lab
xbjxh@live.com

**Xiaodan Liang**
Sun Yat-Sen University
xdliang328@gmail.com

**Liang Lin**
SenseTime Research
linliang@ieee.org

## A. Gradient visualization for the explicit attribute knowledge module

To further investigate the effectiveness of our explicit knowledge module, we use some gradient visualization technique to visualize output features from a "deconvolution approach". More specifically, we adopt the gradient visualization with guided back-propagation [1] on the final layer of the Faster-RCNN and $\mathbf{f}'_a$ of our explicit attribute knowledge module.

Figure 1 shows three representative examples of gradient visualization comparing Faster-RCNN with our explicit attribute knowledge module. Left panels denote the original images and detection results with a visualization threshold of 0.3, the middle panels denote the guided back-propagation saliency graphs and right panels denote the colored guided back-propagation graphs.

From Figure 1, we can see the activation of our module on the middle and right panels is clearer than the Faster-RCNN. For example, from the top comparison, we can find that more areas in the water are activated. Thus more objects are located and recognized in our module. For the middle comparison, the stripes of the zebra in our module are more obvious and we believe this attribute feature of zebra is enhanced by our module. From the last comparison, the features of covered green's leaves are shared and enhanced thus we can see a clear sandwich in the saliency graphs.

## B. Flow charts for Implicit Knowledge Module

Our Implicit Knowledge Module can be found in Figure 2. Suppose $\mathbf{q} = \{q_i\}$ are the input features with the information of some implicit knowledge. $M$ Multiple graphs is used to incorporate different knowledge. We consider a stacked Perceptron for the $m$th knowledge graph as:

$$\hat{e}_{ij}^{(m)} = \varphi_Q(q_i, q_j) = \mathrm{MLP}_Q(q_i - q_j),$$

where $i, j$ is the index of proposed regions, $m = 1, ..., M$. Then each graph is added back a identities matrix $\boldsymbol{I}$ and taken average over $M$:

$$\hat{e}_{ij}^I = \frac{1}{M} \sum_{m=1}^{M} \hat{e}_{ij}^m + \boldsymbol{I},$$

---

[*]Both authors contributed equally to this work.

Figure 1: Gradient visualization with guided back-propagation from the output layer of Faster-RCNN and our explicit attribute knowledge module.

$\hat{\mathcal{E}}^{I} = \{\hat{e}^{I}_{ij}\}$ is then normalized by row and the final output of the module is: $\mathbf{g}' = \hat{\mathcal{E}}^{I}\mathbf{f}\mathbf{W}_{g}$.

Figure 2: Implicit Knowledge Module to learn those knowledge without explicit definitions or being summarized by human. Taking the pairwise differences of the $\mathbf{q}$ as inputs, $M$ different region-to-region graphs are generated by stacked MLP. Then each graph is added back a identities matrix $\boldsymbol{I}$ and taken average over $M$. The output evolved feature $\mathbf{g}'$ is the enhanced feature via graph propagation. Then the output is concatenated to the proposals feature $\mathbf{f}$ to produce final detection results.

## C. Illustration of calculation $Q^A$ and $Q^R$ in the explicit knowledge module

Figure 3 shows an illustration of calculate $\boldsymbol{Q^A}$ and $\boldsymbol{Q^R}$ in the explicit knowledge module.

To construct the ground truth of attribute knowledge $\boldsymbol{Q^A}$, we first get frequent statistics for $K$ attributes and $C$ categories through counting. After normalization, pairwise Jensen–Shannon (JS) divergence between two categories can be measured based on the $C \times K$ frequency distribution table table:

Figure 3: Illustration of how to generate the ground truth attribute class-to-class graph $\boldsymbol{Q^A}$ (top) and relationship graph $\boldsymbol{Q^R}$ (bottom) from the category frequent count.

Figure 4: Improvement of AP over baseline for top 150 infrequent categories.

$$Q_{ij}^A = JS(P_i||P_j) = \frac{1}{2}KL(P_i||\frac{P_i + P_j}{2}) + \frac{1}{2}KL(P_j||\frac{P_i + P_j}{2})$$

where $KL(P_i||P_j)$ is the Kullback–Leibler divergence between two probability distribution $P_i$ and $P_j$. Thus a pairwise $C \times C$ graph $\boldsymbol{Q^A}$ is constructed.

To construct ground truth class-to-class graph $\boldsymbol{Q^R}$ , we first create a $C \times C$ squared matrix $\boldsymbol{R^c}$ with counts from the semantic information. Then, we add the transpose $(\boldsymbol{R^c})^T$ back to $\boldsymbol{R^c}$. Then a column-row normalization is performed to get $\boldsymbol{Q^R}$: $Q_{ij}^R = \frac{R_{ij}^c}{\sqrt{D_{ii}D_{jj}}}$, where $D_{ii} = \sum_{j=1}^C R_{ij}^c$ .

## D. Improvement on infrequent categories

We plot the improvement of our HKRM over the baseline in Figure 4. The bar chart shows the improvement overall AP of the top 150 infrequent categories in VG$_{1000}$ dataset. Solid improvement for our method for those categories can be found. The degraded categories only account for a very small fraction of categories, which only happens 44 from categories with very few samples in VG, biased and noisy annotations of attribute and relationship.

## E. Visualization of different implicit graph

To prove that our implicit module can learn different spatial layouts, 3 random different images are feed into trained implicit module and $M = 10$ learned spatial graphs are extracted. Figure 2 visualizes these spatial graphs where black denotes 0 and red denotes 1. Note that our method learned different kinds of graphs for spatial layout.

## F. Additional Qualitative result comparisons

More qualitative result comparison on VG$_{1000}$ between Faster RCNN and our HKRM can be found in Figure 6. From the comparisons, objects with occlusion, ambiguities and rare category can be detected and localized well by our modules, while the Faster RCNN fail to detect them.

## References

[1] J. T. Springenberg, A. Dosovitskiy, T. Brox, and M. Riedmiller. Striving for simplicity: The all convolutional net. In *ICLR Workshop*, 2015.

Figure 5: Visualization of $M = 10$ implicit graphs from 3 different image.

Figure 6: More qualitative results comparison on $VG_{1000}$ between Faster RCNN and our HKRM. Objects with occlusion, ambiguities and rare category can be detected by our modules.