[Reviews · NeurIPS 2018]

Reviewer 1



Authors observed that most state-of-the-art object detectors, which treat objects individually, can be improved by global reasoning and prior knowledge. The paper proposes a new neural network architecture, Hybrid Knowledge Routed Modules (HKRM), incorporating two types of knowledge, linguistic knowledge (attributes, relationship) and spatial layouts for object detection. The global reasoning/prior knowledge has been proven to be effective for object detection and scene understanding in recent papers including [5,6], Xu et al. "Scene graph generation by iterative message passing, CVPR 2017. Further, as the authors pointed out, for more efficient knowledge representation, matrix (or tensor) factorization can be used for scene understanding as in Hwang et al. Tensorize, Factorize and Regularize: Robust Visual Relationship Learning, CVPR 2018. Authors should include more related works. This paper has many strong points. This paper combines multiple types of knowledge together. The object similarity based on shared attributes was a quite interesting idea like collaborative filtering. The global reasoning is simplified than LSTM or RNN based message passing networks. The proposed methods showed significant improvement against strong baseline (Faster-RCNN). There are a few questions on this paper. It was not clear whether the global reasoning is one-time propagation for each edge. Does it allow the interaction between nodes without edges? If not, can it be considered as global reasoning? How is it sensitive to the order of feature transformation? Authors should provide more discussion on global reasoning. The paper does not have any comparison with other object detectors enhanced by spatial memory [5] global reasoning [6], label relationship [9] and so on. This makes harder to evaluate the utility of the proposed method. The authors said that the proposed method is light-weight. But no data is available to evaluate additional computational cost. If possible, for readers, please compare the computational cost of Faster-RCNN and HKRM. If authors meant that the proposed method is relatively lighter than other knowledge-enhanced methods, relevant data should be provided. Lastly, in the supplementary material, the improvement/degradation of the least frequent 150 categories. Overall, most categories were improved but some of the categories (pencil, manhole) are significantly degraded. Is there any intuition why these classes regressed? Also, it is interesting to see whether the prior knowledge helps object detection for a small set of frequent classes (e.g., 20 VOC classes). Overall, this paper addresses an important problem how to incorporate knowledge in deep neural network specifically for object detection. The improvement is shown against a strong baseline. This work is somewhat incremental.

Reviewer 2



This paper studied large-scale object detection with explicit and implicit knowledge. For explicit knowledge, the weights between different regions are learned according to the weights between different classes in explicit knowledge. For implicit knowledge, the weights between different regions are learned through backpropagation. Strength: - well written paper, easy to read - incorporating knowledge for reasoning is a very important task and attracting growing interest Weakness: -the proposed method seems to be very straightforward. The novelty is marginal - some parts are not clear to me. For example, in line 180, how can you get multiple different graphs? If a different weight is learned for an edge, would it be too much parameter? - some very relevant paper is missing and not cited. Wang et al. Non-local neural networks, which essentially models the implicit relationships between different regions. The authors should also compare this paper

Reviewer 3



This paper is about an object detection method that exploits both implicit and explicit knowledge to increase the semantic consistency of detections in a scene. Starting from one of the current state of art works in object detection (Faster-RCNN) this approach extract object proposal from its RPN, and produce an augmented feature where explicit and implicit knowledge, in the form of feature graphs, is integrated before producing the final classification and bounding box regression. While exploiting the knowledge of spatial, similarity, attributes and context has a long history in the literature (the work cite several of them), this work has the merit of producing adaptive region-to-region graphs and put its foundation on a modern object detection framework that is known to be more reliable than in the past. Moreover the use of Visual Genome and the scene parsing dataset ADE enable the possibility of seeing a marked improvement that was probably not possible in the past. Strengths: + Shows a framework able to integrate explicit and implicit knowledge in a modern framework for object detection such as Faster-RCNN. + Very well written and easy to read. + Enables further studies among this important line of research. Weakness: - The method is tested on ADE and Visual Genome while all previous work on object detection typically use COCO, Pascal VOC 2007 or ILSRVC detection challenge datasets. While they may be considered less challenging than ADE and Visual Genome, it would have been interesting to observe if there is any advantage in using the encoded knowledge in such more simplified dataset. Moreover, it would enable comparison with other state of the art works. - As a continuation of the previous point, ADE is a dataset for Image parsing, which may be biased among different scenes and favour the performance of the proposed method. - The approach is claimed to be "light-weight, general-purpose and extensible by easily incorporating multiple knowledge to endow any detection networks the ability of global semantic reasoning.". Beside Faster-RCNN, since it is an uncommon dataset for object detection, it would be interesting to see more comparisons with other state of the art methods that have better performance than Faster-RCNN. For instance, YOLOv3 and SSD. Such other methods may already have better performance than the proposed method without using any knowledge. Even in such a case, it would not decrease the significance of the paper, but would put more in focus the actual performance that is obtainable by exploiting the knowledge by the proposed method. - The novelty of integrating knowledge is not new. Suggestion: - I would like to suggest the authors to release the source code and/or the trained models. Very minor issues: - row 215: Faser-RCNN -> Faster-RCNN - Faster-rcnn is written differently among the paper. —————————- Thanks to the authors for addressing my concerns. They are all well addressed in the rebuttal.